# A Nucleus-Targeting WT1 Antagonistic Peptide Encapsulated in Polymeric Nanomicelles Combats Refractory Chronic Myeloid Leukemia

**DOI:** 10.3390/pharmaceutics15092305

**Published:** 2023-09-12

**Authors:** Mengting Chen, Xiaocui Fang, Rong Du, Jie Meng, Jingyi Liu, Mingpeng Liu, Yanlian Yang, Chen Wang

**Affiliations:** 1CAS Key Laboratory of Biological Effects of Nanomaterials and Nanosafety, CAS Key Laboratory of Standardization and Measurement for Nanotechnology, CAS Center for Excellence in Nanoscience, National Center for Nanoscience and Technology, Beijing 100190, China; chenmt2019@nanoctr.cn (M.C.); fangxc@nanoctr.cn (X.F.); dur2018@nanoctr.cn (R.D.); mengj@nanoctr.cn (J.M.); liujingyi2019@nanoctr.cn (J.L.); liump2018@nanoctr.cn (M.L.); 2University of Chinese Academy of Sciences, Beijing 100049, China; 3Department of Chemistry, Tsinghua University, Beijing 100084, China

**Keywords:** nucleus-targeting, antagonistic peptide, leukemia, nanomicelle, WT1

## Abstract

Chronic myeloid leukemia (CML) is recognized as a classic clonal myeloproliferative disorder. Given the limited treatment options for CML patients in the accelerated phase (AP) and blast phase (BP), there is an evident need to develop new therapeutic strategies. This has the potential to improve outcomes for individuals in the advanced stages of CML. A promising therapeutic target is Wilms’ tumor 1 (WT1), which is highly expressed in BP-CML cells and plays a crucial role in CML progression. In this study, a chemically synthesized nucleus-targeting WT1 antagonistic peptide termed WIP2W was identified. The therapeutic implications of both the peptide and its micellar formulation, M—WIP2W, were evaluated in WT1^+^ BP-CML cell lines and in mice. The findings indicate that WIP2W can bind specifically to the WT1 protein, inducing cell cycle arrest and notable cytotoxicity in WT1^+^ BP-CML cells. Moreover, subcutaneous injections of M—WIP2W were observed to significantly enhance intra-tumoral accumulation and to effectively inhibit tumor growth. Thus, WIP2W stands out as a potent and selective WT1 inhibitor, and the M—WIP2W nanoformulation appears promising for the therapeutic treatment of refractory CML as well as other WT1-overexpressing malignant cancers.

## 1. Introduction

Chronic myeloid leukemia (CML) is a typical myeloproliferative neoplasm with an annual incidence of approximately 15% among newly diagnosed cases of leukemia in adults [1]. CML is characterized by a balanced genetic translocation involving a fusion of the Abelson murine leukemia (*Abl*) gene from chromosome 9 with the breakpoint cluster region (*Bcr*) gene on chromosome 22, termed the Philadelphia chromosome [2]. The high expression of a constitutively active tyrosine kinase, *Bcr-Abl*, encourages the proliferation and survival of CML cells through diverse molecular mechanisms [3,4,5]. Given the essential role of *Bcr-Abl* tyrosine kinase activity for CML oncogenic transformation, the development of potent and selective inhibitors targeting the *Bcr-Abl* kinase has become a research hotspot [6]. In 2001, imatinib, the first ATP-competitive medication targeting the *Bcr-Abl* kinase, was approved by the Food and Drug Administration (FDA) [7]. Subsequently, second-generation *Bcr-Abl* tyrosine kinase inhibitors (TKIs), such as dasatinib [8], nilotinib [9], and ponatinib [10], were also approved for CML treatment and are considered as the standard of care for CML patients.

CML is a natural triphasic course disease starting with an indolent chronic phase (CP). TKI treatment has dramatically improved the clinical outcomes of patients with chronic CML [11,12]. However, some patients with CP-CML fail to respond to TKIs and progress to the accelerated phase (AP), and eventually to the highly aggressive blast phase (BP) [13]. BP-CML exhibits high resistance to standard induction chemotherapy, resulting in a low response rate (<30%) and a substantially reduced time before the development of drug resistance [14]. The overall survival of patients diagnosed with BP-CML is less than one year [11,15]. At present, BP-CML therapy remains a critical unmet clinical need. Therefore, it is of great significance to develop novel antagonists as alternative therapeutic options for refractory CML.

Wilms’ tumor 1 (WT1) is widely recognized as one of the most prevalent cancer antigens and holds the top rank according to the cancer antigen pilot prioritization: criteria and sub-criteria definitions and weightings, which was published by the National Cancer Institute (NCI) in the United States [16]. In leukemia [17,18] and various types of solid tumors [19,20], high expression of WT1 is often observed. WT1 encodes a zinc finger transcription factor that plays multiple roles in cell proliferation, differentiation, survival, and apoptosis [19,21]. Originally, it was identified as a tumor suppressor gene. However, subsequent evidence suggested that it was, on the contrary, an oncogene. For example, leukemias characterized by high WT1 mRNA and protein levels in leukemic blast cells tend to correlate with a notably worse prognosis, indicating an oncogenic role of WT1 in these leukemic cells [22]. The prognostic value of WT1 has been validated in CML patients, with high expression levels detected in relapsed CML and low expression levels found in complete remission [18,23]. In addition, the expression of WT1 in CML exhibits an incremental pattern along with disease progression. Specifically, WT1 levels are relatively low during the chronic phase but show a significant increase as the disease progresses to a blast crisis [23,24]. This suggests that WT1 overexpression plays a role in the progression of CML. Therefore, there is an urgent need to develop novel and potent inhibitors to effectively treat advanced CML driven by WT1. Currently, drug development strategies targeting WT1 primarily focus on cancer vaccines, and there is a lack of antagonistic drugs specifically targeting WT1. Targeting and interfering with the function of this “undruggable” transcriptional regulator remains a challenge for small molecule inhibitors [25]. In addition, traditional therapeutic agents have limited access to the WT1 protein in the nucleus. Hence, the development of novel WT1-targeting drugs is a challenging endeavor.

Recent progress in pharmaceutical technologies has led to the emergence of therapeutic biologics including peptides, antibodies, and recombinant proteins [26,27]. Among them, therapeutic peptides exhibit several advantages, such as the ability to target a wide range of intercellular and intracellular antigens, chemical synthetic feasibility, low immunogenicity, high activity, and binding selectivity [27,28]. Peptides have demonstrated promising potential in inhibiting tumor progression [29], primarily by targeting and suppressing abnormally expressed oncoproteins [30,31]. For example, our previous study identified a series of novel antagonistic peptides targeting CXCR4 [30,32], CD123 [31], and EZH2 [33]. However, conventional peptides often suffer from drawbacks such as limited membrane permeability, metabolic instability, and restricted tissue residence, ultimately resulting in limited bioavailability [34].

Several studies have reported that nanoparticle-based therapeutics have the potential to enhance the effectiveness of anticancer treatments while minimizing their side effects [35,36,37]. These nanoscale systems can penetrate tumor tissues, facilitate active internalization, and exhibit preferential accumulation in tumors through enhanced permeation and retention (EPR) effects [38]. Various types of nanocarriers, such as liposomes [39,40], polymer micelles [41], and solid lipid nanoparticles [42], constitute a substantial proportion of nanotherapeutics currently undergoing clinical trials [37]. Polymer micelles have emerged as a prominent type of lipid-based nanocarriers, primarily due to their drug preparation method, which enables cost-effective and highly reproducible manufacturing on a large scale [37]. Polymeric micelles are composed of a hydrophobic core stabilized by a hydrophilic corona [41], exhibiting an amphiphilic structure that facilitates the efficient incorporation of peptides. The corona layer of the micelle provides effective steric protection for the peptides. The utilization of polymeric micelles for the encapsulation of peptides often leads to prolonged circulation time, improved biodistribution, and preserved biological activity. In our previous studies, we conducted experiments on polymeric micelles composed of PEG-PE (Distearoyl-*sn*-glycero-3-phosphoethanolamine-n-[methoxy(polyethylene glycol)-2000]) using a one-step self-assembly procedure [31,32,33,43]. This procedure enables the encapsulation of water-soluble small molecules and anticancer peptides. Compared with free peptides, the peptide-loaded PEG-PE micelles exhibited enhanced antitumor capabilities at cellular and tumor-bearing mouse levels [31,33]. Moreover, the PEG-PE copolymer has been approved by the Food and Drug Administration (FDA) as a stealth material for Doxil^®^, which is a liposomal formulation of doxorubicin. Additionally, PEG-PE micelles possess a remarkably low critical micelle concentration (CMC) of 1 µM, ensuring structural integrity during circulation in the bloodstream. Therefore, PEG-PE micelles hold great promise as a nanoformulation for anticancer peptides.

Since the WT1 protein is primarily localized in the nucleus, it is crucial to develop nucleus-targeting drug delivery strategies to enhance drug retention and efficacy. In this regard, the HIV-1 trans-activator of transcription (TAT) peptide is employed for delivering drugs to the nuclei of tumor cells. TAT possesses the ability to actively penetrate cell membranes and nuclear membranes through the nuclear pore [44]. To ensure optimal uptake of the cargo and its subsequent delivery into the nucleus of cancer cells, conjugation of the cargo with both the TAT peptide and the nuclear localization sequence is necessary [44,45]. Our previous studies have demonstrated the indispensability of TAT-mediated transduction for facilitating the transport of nucleus-targeting antagonistic peptides across membranes and into the nucleus [33]. However, the intrinsic non-specificity of TAT limits its application in vivo, arising from its cationic charge in a biological environment [46]. Therefore, it is necessary to implement measures to protect TAT conjugated with anticancer agents from external exposure until they reach the tumor site. With the development of nanotherapeutics, nanocarriers are utilized to efficiently deliver therapeutic agents conjugated with TAT [47].

Given the clinical demand for WT1-driven advanced CML therapy, as well as the ongoing challenges of delivering therapeutic peptides to the nuclei of tumor cells in vivo, we have designed a potential strategy for refractory CML therapy, which involves the encapsulation of a TAT-mediated nucleus-targeting WT1 antagonistic peptide with PEG-PE. This chemically synthesized antagonist, named WIP2W, comprises the WT1 targeting sequence (WWGGDQRRSWGRRRPDRR) and the TAT sequence (YGRKKRRQRRR). Experimental results have confirmed that WIP2W can specifically bind to intracellular WT1 protein, and its encapsulation in nanomicelles enables effective nucleus delivery. Leveraging EPR effect, the nanomicellar form of WIP2W, referred to as M—WIP2W, exhibits enhanced intra-tumoral accumulation in WT1^+^ CML mice as compared to free WIP2W, leading to effective inhibition of tumor growth. In conclusion, this study demonstrates the promising potential of M—WIP2W as a WT1 inhibitor for refractory CML treatment. Furthermore, this work also illustrates the potential application of WIP2W and M—WIP2W in other WT1-overexpressing malignant cancers.

## 2. Materials and Methods

### 2.1. Chemicals and Reagents

PEG-PE, Distearoyl-*sn*-glycero-3-phosphoethanolamine-n-[methoxy (polyethylene glycol)-2000], was procured from Jiangsu Southeast Nanomaterials Co., Ltd. (Huaian, China). Imatinib mesylate powder was purchased from MedChemExpress (Shanghai, China). Streptavidin magnetic beads were acquired from Jiangsu BEAVER Biomedical Co., Ltd. (Suzhou, China). The anti-WT1 and GAPDH antibodies were obtained from Abcam (Boston, MA, USA) and Proteintech (Rosemont, IL, USA). Secondary antibodies were procured from Cell Signaling Technology (Danvers, MA, USA). Both the protein extraction reagent kits and protease inhibitor cocktails were bought from Thermo Fisher Scientific (Waltham, MA, USA). Phosphate-buffered saline (PBS, pH 7.4) was purchased from Gibco (Waltham, MA, USA).

### 2.2. Chemically Synthesized Peptides

The peptide WIP2W (WWGGDQRRSWGRRRPDRRYGRKKRRQRRR) was chemically synthesized by Guoping Pharmaceutical Co., Ltd. (Hefei, China) with a purity of >98% via solid-phase peptide synthesis (SPPS). According to the specific requirements of the experiments, biotin and fluorescein isothiocyanate (FITC) were conjugated with the N-terminal of WIP2W to obtain biotin-WIP2W and FITC-WIP2W. A molecule of Acp (6-aminocaproic acid), also called an alkyl spacer, is inserted between FITC and WIP2W for the synthesis of FITC-WIP2W. The molecular weight identification of synthetic peptides including WIP2W, biotin-WIP2W, and FITC-WIP2W is shown in the Appendix A. In this study, unlabeled WIP2W, FITC-WIP2W, and biotin-WIP2W powders were completely solubilized in PBS (pH 7.4) in advance, at specific concentrations.

### 2.3. Cell Lines and Animals

Human chronic myeloid leukemia cell line K562 and normal hepatocyte cell line L02 were cultivated in media from Gibco (USA), enriched with 10% fetal bovine serum (FBS) and 1% penicillin/streptomycin (PS) and maintained at 37 °C. Female BALB/c nude mice were obtained from SPF Biotechnology Co., Ltd. (Beijing, China).

### 2.4. Immunoblot Assay

The WT1 expression levels in K562 cells (overall, nuclear, and cytoplasmic) and L02 cells (overall) were assessed using Western blot analysis. In brief, proteins from the cultured cells were extracted using lysis buffers supplemented with protease inhibitors. Equal amounts of the protein samples were then separated and detected with appropriate antibodies, namely anti-WT1 (1:1000, *v*/*v*) and GAPDH (1:20,000, *v*/*v*). The resulting immunocomplexes were visualized on a gel imaging system from Bio-Rad (Hercules, CA, USA), using the Immobilon ECL Western detection reagents from Solarbio (Beijing, China).

### 2.5. Intracellular Localization of WT1 and WIP2W

To ascertain the intracellular localization of WT1, immunofluorescence was employed. K562 cells underwent a process that included collection, fixation, permeabilization, blocking, and labeling with the anti-WT1-Alexa Fluor 647 antibody (1:200, *v*/*v*). Following nuclear staining with DAPI, K562 cells were transferred to confocal dishes. A confocal laser scanning microscope (Carl Zeiss, Oberkochen, Germany) was used to capture the fluorescence images.

The precise localization of WIP2W was identified using the same microscope. K562 cells, after being exposed to FITC-WIP2W (10 μM) at 37 °C for 3.5 h, were gathered and stained with Hoechst33342. They were then moved to confocal dishes, where fluorescence data were captured.

### 2.6. Pull-Down Assay

Proteins from different localizations were harnessed for peptide–protein interaction studies [33,48]. Initially, biotin-WIP2W and streptavidin-coated magnetic beads were mixed in PBS and shaken at 25 °C for 1 h. The bound WIP2W beads were magnetically separated, discarding the unattached WIP2W molecules in the leftover solution. The combined protein extracts from the whole cell, nucleus, and cytoplasm were then exposed to the WIP2W beads for another 2 h at 25 °C. Finally, the bead eluates were probed using an anti-WT1 antibody.

### 2.7. Cytotoxicity Assay

The cytotoxicity of WIP2W and M—WIP2W against cells was represented in the suppression of cell viability. K562 (1.5 × 10^4^/well) and L02 (8 × 10^3^/well) cells were seeded in 96-well plates and incubated with gradient concentrations of peptides or micelles in complete mediums. After 24 or 48 h of incubation, each well was spiked with CCK-8 agent (10 μL) for another 2 h. Absorption was detected by a spectrophotometer at 450 nm (Molecular Devices, San Jose, CA, USA), and IC_50_ values were calculated by curve fitting.

### 2.8. Cell Cycle Analysis

K562 cells were incubated in the absence and presence of WIP2W (10 μM) for 24 h. Subsequently, these cells were collected and fixed with 75% cold ethanol overnight, followed by treatment with propidium iodide (PI) and RNase A. Then, the cell fluorescence was examined using flow cytometry, and the cell cycle profiles were analyzed using FlowJo 10.

### 2.9. RNA Sequencing and Differentially Expressed Gene Analysis

K562 cells were incubated in the absence and presence of WIP2W (10 μM) for 24 h. These cells were harvested, and the total RNA was then collected for subsequent RNA sequencing (RNA-Seq) tests. We sequenced the library using the Illumina NovaSeq 6000 platform (Illumina, San Diego, CA, USA), which generated 150 bp paired-end reads. The processed clean reads were aligned using HISAT2, and FPKM values were calculated for each gene. Differentially expressed genes (DEGs) were subsequently subjected to the Kyoto Encyclopedia of Genes and Genomes (KEGG) pathway enrichment analysis.

### 2.10. Preparation of M—WIP2W and Empty PEG-PE Micelles

The one-step self-assembly method previously described is suitable for PEG-PE and the water-soluble WIP2W [31,32,33,49]. Prior to experimentation, each powder was dissolved in PBS (pH 7.4) at specified concentrations. Solutions of PEG-PE and WIP2W were mixed in molar ratios varying from 0.5 to 20 equivalents of PEG-PE per WIP2W unit. The mixtures were then gently heated between 50 °C and 55 °C for 30 min to encapsulate the WIP2W. The resulting M—WIP2W solution was stored at 4 °C. In the absence of WIP2W, empty micelles were produced under the same conditions.

### 2.11. Characterization of M—WIP2W and PEG-PE Micelles

The zeta potential, average hydration radius, and PDI of M—WIP2W (comprising 20 μM PEG-PE and 6.7 μM WIP2W) and 20 μM PEG-PE micelles were evaluated using a nanoparticle analyzer (Malvern Instrument Ltd., Malvern, UK). The data are presented as the mean ± standard deviation (SD). The structures of M—WIP2W and PEG-PE micelles were examined using a transmission electron microscope (Hitachi, Ltd., Tokyo, Japan).

### 2.12. Encapsulation Efficiency of Micelles to WIP2W

The ability of PEG-PE micelles to encapsulate WIP2W was assessed using an ultrafiltration method. Solutions of FITC-WIP2W-loaded micelles (with PEG-PE concentrations of 5, 10, 30, 50, 100, and 200 μM, and FITC-WIP2W at 10 μM) were placed into an ultra-centrifugal filter (MWCO 100,000, Millipore, Burlington, MA, USA). They were then centrifuged at 12,000× *g* for 30 min. The fluorescence intensity of the filtrate was subsequently measured spectrophotometrically. A concentration-to-fluorescence intensity relationship was then derived to determine the amount of unencapsulated FITC-WIP2W in the filtrate. From this, the encapsulation efficiency (*EE*%) of PEG-PE micelles for WIP2W can be calculated, as shown below.
*EE*% = incorporated WIP2W/initial WIP2W added × 100%(1)

### 2.13. Cellular Uptake of WIP2W

The efficiency of WIP2W and M—WIP2W uptake was represented by both the percentage of positive cells and the mean fluorescence intensity (MFI). K562 cells were treated with FITC-WIP2W at concentrations of 0.05, 0.5, and 5 μM, and M—WIP2W (consisting of 15 μM PEG-PE and 5 μM FITC-WIP2W) at 37 °C for 2 h. The percentage of positive cells and their mean fluorescence intensity were then analyzed using a flow cytometer, with each condition tested in triplicate (*n* = 3).

### 2.14. In Vivo Biodistribution of WIP2W and M—WIP2W

In this study, a WT1^+^ BP-CML xenograft mouse model was developed. K562 cells (1 × 10^7^/animal) were injected subcutaneously into the flanks of 5-week-old female BALB/c nude mice to generate primary BP-CML tumors. Once the tumors became noticeable, the mice were euthanized, and tumor cells were harvested. These cells were then reintroduced subcutaneously into a new set of hosts (1 × 10^7^/animal) for further therapeutic investigations. The biodistribution of WIP2W and M—WIP2W in vivo was examined by administering FITC-WIP2W (at a dose of 30 mg/kg WIP2W) and M—FITC-WIP2W (64 mg/kg PEG-PE and 30 mg/kg WIP2W) to BP-CML mice. Four hours post-administration, the mice were euthanized. Their tumors and primary organs were then extracted and assessed for fluorescence using IVIS (in vivo imaging system) equipment from PerkinElmer, USA.

### 2.15. In Vivo Antitumor Efficiency of WIP2W and M—WIP2W

Xenografted WT1^+^ BP-CML mice were utilized to evaluate the anti-tumor effects of WIP2W and M—WIP2W. By the 4th day post-CML cell implantation, tumors in the secondary mice had grown to an average size of about 100 mm^3^. These tumor-bearing mice were then randomly assigned to one of four groups (*n* = 5 per group). For the subsequent 14 days, the groups received daily treatments as follows: control (PBS, pH 7.4), imatinib (50 mg/kg, intraperitoneal injection), WIP2W (30 mg/kg, subcutaneous injection), and M—WIP2W (PEG-PE: 64 mg/kg, WIP2W: 30 mg/kg, subcutaneous injection). The subcutaneous tumor volume (mm^3^) and body weight (g) were recorded three times per week, and the tumor volume can be computed according to the following equation:Tumor volume (mm^3^) = L × W × W/2(2)
where “L” and “W” denote the maximum and minimum dimensions, respectively. On day 19, the mice were slaughtered, and the retrieved major organs and tumors were stained with hematoxylin and eosin (H&E), WT1, and Ki67, respectively.

### 2.16. Statistical Analysis

All quantitative data are represented as mean ± SD, derived from a minimum of three independent experiments. The statistical significance of the results was determined using an unpaired Student’s *t*-test for comparisons between two groups and a one-way ANOVA for multiple group comparisons. Levels of significance are denoted by *p*-values (ns: No significance, * *p* < 0.05, ** *p* < 0.01, and *** *p* < 0.001).

## 3. Results

### 3.1. Synthetic WIP2W Specifically Binds to the WT1 Protein

The K562 cell line, sourced from a CML patient during a blast crisis, served as a representative model of the BP-CML cell line [50]. To pioneer peptides that antagonize WT1, we initially verified the WT1 expression in K562 cells. Immunoblot analysis of total protein lysates from K562 is displayed in Figure 1A (left lane). The findings highlighted a prominent WT1 protein expression in K562 cells, aligning with prior research [51]. Both nuclear and cytoplasmic protein extracts underwent immunological assays to discern WT1’s subcellular distribution in K562 cells. The WT1 protein was found in the nucleus (Figure 1A, middle lane) and cytoplasm (Figure 1A, right lane), predominantly residing in the nucleus. An immunofluorescence assay further validated the intracellular positioning of WT1 in K562 cells, with pronounced red fluorescence signals pinpointed in the nucleus, corroborating the nuclear location of the WT1 protein (Figure 1B), consistent with the Western blot results.

WT1 antagonistic peptides offer a promising avenue for developing WT1-targeting therapeutics in leukemia. Our recent study demonstrated that TAT-mediated transduction is necessary for the transmembrane transport of the nucleus-targeting antagonistic peptide [33]. Considering the characteristics of the WT1 protein, mainly localized in the nucleus, the designed WT1 antagonistic peptides in this study contained two functional regions: the WT1-targeting peptide sequence and the TAT sequence. By screening cell cultures from a series of de novo-designed peptides targeting various segments of WT1 protein (Appendix A), a well-performing WT1 antagonist (WIP2W) was obtained. For WIP2W, we anticipated that the TAT sequence of WIP2W would adequately guide the WT1-targeting peptide sequence to the K562 cell nucleus where WIP2W would then identify the WT1 protein.

To verify this hypothesis, the uptake of WIP2W by K562 cells was gauged via flow cytometry. The cellular uptake of WIP2W exhibited concentration dependency, signifying WIP2W’s successful entry into K562 cells (Figure 1C). Additionally, fluorescence imagery affirmed that post-incubation with FITC-tagged WIP2W, the majority of the green fluorescence was pinpointed in K562 cells’ nuclei (Appendix A). These data imply that the TAT sequence [52,53] facilitates WIP2W’s passage across cellular and nuclear membranes, culminating in nuclear residency and target protein interaction. Additionally, a pull-down assay was conducted to validate the interaction between WIP2W and the WT1 protein. As depicted in Figure 1D, streptavidin magnetic beads were modified with biotin-labeled WIP2W through a biotin–streptavidin interaction. The functionalized beads were then incubated with protein lysates, including the nuclear and cytoplasmic proteins from K562 cells. Afterward, the proteins captured by WIP2W underwent immunoblot examination using an anti-WT1 antibody. The outcomes revealed that WIP2W could specifically bind to the WT1 protein in both the nucleus and cytoplasm, underscoring the recognition specificity between WIP2W and WT1 (Figure 1E).

### 3.2. Anti-BP-CML Effect of WIP2W In Vitro

Evidence has shown that high WT1 expression in leukemia may contribute to the maintenance of malignant phenotypes via various molecular mechanisms [19,21]. We measured the inhibitory activity of WIP2W against WT1 in vitro, which ultimately reflected cytotoxicity. The cytotoxicity of WIP2W on different cell lines for 24 h is shown in Figure 2A. WIP2W displayed pronounced concentration-dependent cytotoxicity in K562 cells; the cell viability dropped to <10% after incubation with 20 and 40 μM WIP2W (orange line). The estimated WIP2W concentration that inhibited 50% K562 cell proliferation (IC_50_) was 9.7 μM. Notably, WIP2W exhibited no apparent cytotoxicity in human normal hepatic cell line L02, even when incubated with 40 μM WIP2W for 24 h (black line). This may be related to the extremely low level of WT1 in L02 cells (Figure 2A). The selective cytotoxicity of WIP2W on WT1-overexpressing cancer cells and WT1-low-expressing normal cells, suggests the specific WT1 antagonistic effect and great biocompatibility of WIP2W. Previous studies suggested that WT1 inhibitors had the potential to halt cell cycle progression. Given this, we employed flow cytometry to explore the impact of WIP2W on the progression of the K562 cell cycle (Figure 2B,C). Administering 10 μM WIP2W effectively increased the G_0_/G_1_ phase compartment of cell cycle distribution from 24.1% to 32.4% while diminishing the G_2_/M phase compartment from 22.4% to 14.1%, relative to the control group. These results indicated that WIP2W could impede K562 cell cycle progression in the G_0_/G_1_ phase.

Subsequently, we investigated the anti-BP-CML effects of WIP2W at a genetic level utilizing RNA sequencing. Hierarchical clustering analysis of the differentially expressed genes (DEGs) showcased genetic alterations post-WIP2W treatment in comparison to the untreated group (Figure 3A). Moreover, the Kyoto Encyclopedia of Genes and Genomes (KEGG) pathway enrichment analysis revealed significant shifts in certain biological processes after WIP2W administration, including the receptor signaling pathway, PI3K-Akt signaling pathway, ECM–receptor interaction, metabolic processes, and the TNF signaling pathway, among others (Figure 3B). In-depth analysis of the DEGs that exhibited significant changes in K562 cells treated with WIP2W showed an elevation of tumor suppressor genes, including OSCP1, and genes related to cell apoptosis such as FXYD6, NPTX1, and GPR142. Conversely, genes associated with cell adhesion and migration, such as LAMC2, were notably reduced (Figure 3C). These genetic alterations might be responsible for the inhibition of K562 cells induced by WIP2W.

### 3.3. Preparation, Characterization, and Cytotoxicity of M—WIP2W

Peptides have attracted attention in drug discovery due to their therapeutic efficacy. Yet, their instability in circulation and unsatisfactory biodistribution pose significant challenges for in vivo peptide delivery. Nanoparticles made from biodegradable polymers offer a promising avenue for peptide delivery, owing to their increased solubility, stability, circulating life, and pharmacokinetic properties [32,33,54,55]. Inspired by this, WT1 antagonistic peptide WIP2W was assembled with amphiphilic PEG-PE to enhance its bioavailability. First, empty micelles and the nanomicellar form of WIP2W (namely M—WIP2W) were fabricated by a self-assembly method [31,32,33]. Within M—WIP2W, the electrostatic interactions between PEG-PE and WIP2W ensure the stable encapsulation of the drug in the micelle. Briefly, in neutral solutions, the negatively charged PEG-PE and the positively charged WIP2W should attract each other due to their contrasting isoelectric points—estimated at 12.35 for WIP2W and 5.93 for the PEG-PE polymer.

As shown in Appendix A, the encapsulation efficiency (*EE*%) of WIP2W within PEG-PE micelles was influenced by the molar ratio of PEG-PE to WIP2W. As this ratio increased from 0.5:1 to 20:1, the *EE*% of WIP2W in micelles saw a steady rise until stabilizing. At a 3:1 molar ratio (PEG-PE: WIP2W), the EE% was approximately 72% and exceeded 95% when the molar ratio surpassed 10. Considering the potential vehicle-induced cytotoxicity, the toxicity of PEG-PE alone was examined in K562 cells (Appendix A). The results showed that cell viability remained unaffected at PEG-PE concentrations less than 30 μM. However, viability decreased to 77% at a PEG-PE concentration of 60 μM. Considering *EE*% and vehicle-induced toxicity, we opted for a constant 3:1 molar ratio of PEG-PE to WIP2W for subsequent investigations.

The physicochemical properties of empty PEG-PE micelles (20 μM) and M—WIP2W (PEG-PE: 20 μM, WIP2W: 6.7 μM) were analyzed using TEM and DLS, including the morphology, average hydration radius, zeta potential, and polydispersity index (PDI). As depicted in Figure 4A,B, both empty micelles and M—WIP2W exhibited a near-spherical shape with an average diameter of around 23–24 nm. Table 1 summarizes these physicochemical characteristics. The diameter remained relatively unchanged post-WIP2W encapsulation. Empty micelles registered a hydrodynamic size of 23.3 ± 2.8 nm and a zeta potential of −4.0 ± 0.6 mV. Meanwhile, M—WIP2W displayed a hydrodynamic size of 23.9 ± 3.2 nm and a zeta potential of 5.3 ± 0.3 mV. The increased zeta potential indicated the successful encapsulation of WIP2W within the PEG-PE micelles. In summary, micellar WIP2W (M—WIP2W) was easily fabricated through a simple procedure.

To evaluate the uptake ability of WIP2W in the presence and absence of nanomicelles, K562 cells were incubated with either free FITC-WIP2W (5 μM) or M—FITC−WIP2W (PEG-PE: 15 μM, FITC-WIP2W: 5 μM) for 2 h at 37 °C. The flow cytometry results, presented in Figure 4C, demonstrate that K562 cells internalized a marginally greater amount of M—WIP2W compared to the free WIP2W. This suggests that the PEG-PE micelles enhance the delivery of WIP2W into K562 cells. Such observed enhancement in cellular uptake may be attributed to the optimized membrane fluidity triggered by amphiphilic PEG-PE insertion [56]. Subsequently, the relative cell viability was analyzed post-incubation with gradient concentrations of WIP2W and M—WIP2W. As indicated in Figure 4D, the cytotoxic impacts of WIP2W and M—WIP2W were comparable, and M—WIP2W demonstrated a slightly elevated cytotoxicity compared to free WIP2W within a concentration range of 5–20 μM. Notably, M—WIP2W exhibited minimal cytotoxic effects on L02 cells, highlighting the selective cytotoxic nature of M—WIP2W (Appendix A).

### 3.4. Anti-BP-CML Effect of WIP2W and M—WIP2W In Vivo

To investigate the biodistribution of WIP2W in the absence and presence of PEG-PE micelles in vivo, we established a K562 cell-bearing BALB/c nude mouse model, termed WT1^+^ BP-CML mice. The biodistribution was investigated by detecting the fluorescence in the excised tumors and major organs (including the heart, liver, spleen, lung, and kidney) using the IVIS (in vivo imaging system). Four hours after a single subcutaneous injection of PBS, FITC-WIP2W, or M—FITC-WIP2W, the tumors and organs from euthanized mice were imaged for fluorescence (Figure 5A). The PBS group served as a control (left lane). As anticipated, the intra-tumoral accumulation of M—WIP2W in the BP-CML mice surpassed that of free WIP2W, possibly due to EPR effect [31,32,33,54,57]. In addition, in FITC-WIP2W-treated mice, free WIP2W was predominantly distributed in the liver and kidneys (middle lane), hinting that WIP2W introduced subcutaneously was cleared through the hepatic and renal metabolic pathways. By contrast, for the M—WIP2W-treated mice, a pronounced fluorescence was observed in the liver, with minimal fluorescence in the kidneys (right lane), possibly due to the size effect. Typically, nanoparticles’ metabolic pathways hinge on their size [58,59]. Nanoparticles exceeding 6 nm in diameter are often processed by the liver and spleen, while those smaller than 6 nm are processed by the kidneys [58]. In addition, if nanoparticles are smaller than 10 nm, they are quickly cleared by the kidneys and do not easily accumulate at the tumor site through EPR effect [60]. In our study, M—WIP2W, approximately 20 nm in size, showed promise with regard to efficient distribution in the body.

To delve deeper into the in vivo anti-tumor efficacy of WIP2W and M—WIP2W, WT1^+^ BP-CML mice received daily treatments of PBS, imatinib, WIP2W, and M—WIP2W for two weeks. Imatinib, currently the first curative for patients with CML, was chosen as the positive control [7]. When the tumor volume approached 100 mm^3^, BP-CML mice were administered daily subcutaneous injections of PBS, WIP2W (30 mg/kg), and M—WIP2W (WIP2W: 30 mg/kg; PEG-PE: 64 mg/kg), respectively. Imatinib was given intraperitoneally at 50 mg/kg daily. During the succeeding 14 days, the tumor volume (mm^3^) and body weight (g) of BP-CML mice were recorded thrice weekly. On the 15th day post-administration, the mice were euthanized. The excised tumors underwent immunohistochemical analysis using an anti-WT1 antibody. H&E and Ki67 staining of tumors and major organs were also conducted.

Compared to the control group, imatinib, WIP2W, and M—WIP2W exhibited potent anti-tumor activity in the BP-CML mice (Figure 5B). Remarkably, M—WIP2W outperformed free WIP2W in mitigating BP-CML progression, an outcome attributed to the enhanced intra-tumoral peptide accumulation. Body weight tracking revealed no significant weight loss in the WIP2W and M—WIP2W groups compared to the control and imatinib groups, hinting at the biocompatibility of both treatments (Figure 5C). H&E staining of the tumor tissues identified pronounced cell death in the WIP2W and M—WIP2W groups (Figure 5D), suggesting their tumor-inhibiting potency. Additionally, WIP2W and M—WIP2W treatment significantly diminished WT1 protein levels in the tumor tissues relative to controls (Figure 5D,E), underscoring WIP2W’s antagonistic effect. The distinct interaction between WIP2W and WT1 might induce structural changes in the WT1 protein, potentially curbing the full expression of WT1’s function.

To verify the biocompatibility of WIP2W and M—WIP2W, we also conducted H&E staining of major organs. Both WIP2W and M—WIP2W exhibited minimal systemic toxicity in mice, as shown in Figure 6A. Contrastingly, liver damage was evident in the imatinib group, suggesting a potential side effect of continuous imatinib administration. Given that leukemia is notably aggressive and frequently metastasizes to the spleen [61], we explored the efficacy of imatinib, WIP2W, and M—WIP2W in inhibiting tumor metastasis. After various treatments, spleen samples from BP-CML mice were collected and subjected to Ki67 staining (Appendix A). In both the control and imatinib groups, the intensity of Ki67 staining (Figure 6B) exhibited pronounced fluorescence in the spleen, indicating the splenic infiltration of CML cells. Encouragingly, the mice receiving WIP2W and M—WIP2W treatments showed negligible signs of metastatic CML cells in the spleen, suggesting the effective metastasis prevention capabilities of WIP2W and M—WIP2W. Collectively, these findings suggest that WIP2W and M—WIP2W have the potential to be effective and safe candidates for treating refractory CML and other malignancies with elevated expression of WT1.

## 4. Discussion

In CML, the influence of WT1 stems from its robust transcriptional control of genes associated with cell growth, metabolism, differentiation, and apoptosis. Interventions targeting WT1 can induce cell cycle arrest and apoptosis. Previous research found that WT1 inhibitors could impact cell cycle progression in myeloid leukemia. For example, curcumin, a WT1 inhibitor, can induce cell death and G_2_/M phase arrest [51]. Similarly, WT1-specific RNA interference led to an increase in G_0_/G_1_-phase cells and a decrease in S-phase cells [51]. In this study, the WIP2W peptide halted the cell cycle in the G_0_/G_1_ phase and reduced WT1 protein levels in tumor tissues. The mechanism of WIP2W is partially related to cell cycle arrest, and other mechanisms, including apoptosis, are under investigation.

Nucleus-targeting WT1 antagonist peptides offer a promising strategy for WT1^+^ BP-CML therapy. Our previous studies demonstrated the necessity of TAT-mediated transduction for the transportation of nucleus-targeting antagonistic peptides across membranes and nuclei [33]. In this work, we found that the TAT sequence of WIP2W would adequately guide the WT1-targeting peptide sequence to the K562 cell nucleus where WIP2W would then identify the WT1 protein. Until now cationic cell-penetrating peptides (CPPs) have been employed to bypass biological barriers and deliver cargo into cancer cells noninvasively [52,62]. By combining subcellular-targeting CPPs with targeting peptides, it is possible to deliver peptides to specific cell organelles in a precise manner [53].

Peptides present instability in circulation and unsatisfactory biodistribution challenges for in vivo application. To enhance the bioavailability and tissue residence of peptides, we employ PEG-PE nanomicelles for loading and delivering WIP2W. In tumors with high WT1 expression, M—WIP2W showed superior antitumor effects compared to free peptides. This increased efficacy was largely due to M—WIP2W’s enhanced tumor accumulation capabilities. Beyond particle size, the surface characteristics and shape of nanoparticles can also influence their biodistribution [63,64].

Recent clinical findings suggest that many CML patients develop resistance to imatinib with continued use [65,66]. Approximately 70% of relapsing CML patients have mutations in the *Bcr-Abl* kinase domain, with some even impacting the imatinib binding site, thereby instigating drug resistance [66]. Notably, BP-CML is often unresponsive to conventional induction chemotherapy [14]. WT1 is crucial to oncogenic survival signaling downstream of *Bcr-Abl* in leukemia and counteracts the cytotoxic effects of TKIs like imatinib [67]. Our study found that while BP-CML mice had a limited response to imatinib, they responded well to M—WIP2W therapy, suggesting its potential as a complementary treatment alongside TKIs.

The spleen, a primary site for extramedullary hematopoiesis (EMH), is commonly implicated in hematological malignancies [68]. For example, leukemic cells frequently infiltrate the spleen in CML cases [69]. Our findings showed that mice treated with WIP2W and M—WIP2W had minimal metastatic CML cells in the spleen, suggesting these treatments effectively prevent tumor spread. Conversely, PBS- and imatinib-treated groups exhibited significant splenic CML infiltration. Moreover, leukemia often metastasizes to the bone marrow [61], guiding our subsequent evaluations of metastatic inhibition.

## 5. Conclusions

The tumor antigen WT1 located in the nucleus is overexpressed in both BP-CML and a variety of solid tumor cells, which was associated with drug resistance and poor prognosis. Currently, there is no WT1-targeting antagonistic drug available, which may be attributed to the multiple technical challenges in intracellular antigen-targeting drug development. In our study, we tackled these long-standing challenges with the proposed antagonistic peptide sequence enabled by nanocarriers. The results demonstrated that the novel and chemically synthesized antagonistic peptide WIP2W can specifically bind to the intracellular WT1 protein, and its micellar formulation (M—WIP2W) can significantly enhance cell death and inhibit the tumor growth of WT1^+^ BP-CML mice. Therefore, our study may provide a highly promising therapeutic strategy to improve clinical treatments for refractory CML as well as other WT1-overexpressing malignant cancers. Also, targeting intracellular antigens is a highly strategic drug design approach that will greatly enrich the variety of cancer antigens. This study illustrates the feasibility of developing intracellular antigen-targeting delivery of therapeutic peptides for cancer therapy.

## Figures and Tables

**Figure 1 pharmaceutics-15-02305-f001:**
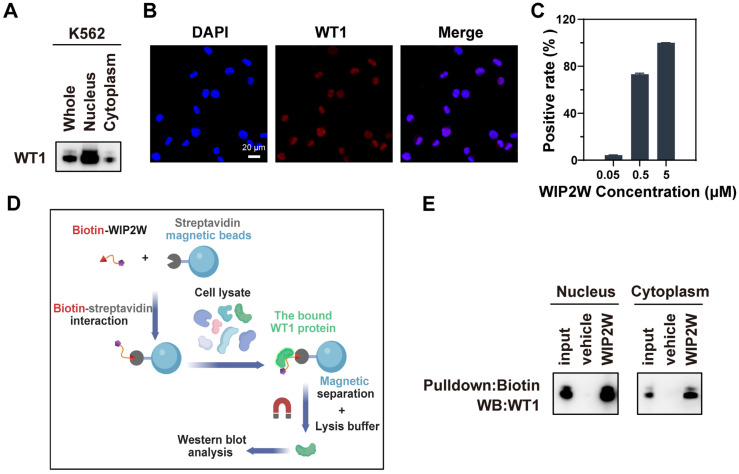
WIP2W could specifically bind to the WT1 protein overexpressed in K562 cells. (**A**) WT1 subcellular expression was evaluated via immunoblot analysis using an anti-WT1 antibody. (**B**) Intracellular localization of WT1 protein in K562 cells was examined via immunofluorescence assay. The blue fluorescence signifies the nucleus. The red signal represents the WT1 protein. The right figure displays the co-localization (scale bars: 20 μm). (**C**) Cellular uptake of FITC-WIP2W (0.05, 0.5, and 5 μM) by K562 cells was determined using flow cytometry after incubation at 37 °C for 2 h. Results from triplicate experiments were analyzed using GraphPad Prism 8. Data are presented as mean ± SD. (**D**) Schematic illustration of the pull-down assay. Created with BioRender.com. (**E**) Eluates obtained from the pull-downs (nucleus and cytoplasm protein) were subjected to Western blot assay using an anti-WT1 antibody. No WIP2W-modified magnetic beads were incubated with cell lysates and served as vehicle control. Cell lysate (5% input) served as a positive control.

**Figure 2 pharmaceutics-15-02305-f002:**
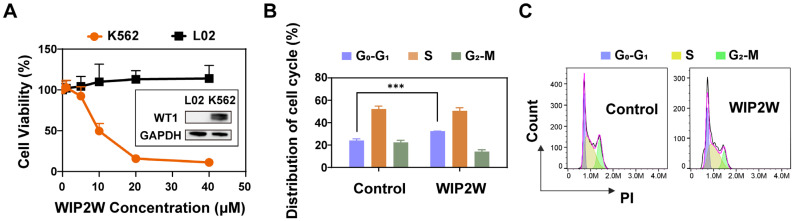
WIP2W effectively induced cell death and cycle arrest in K562 cells. (**A**) Cytotoxicity of WIP2W on different cell lines. K562 cells (1.5 × 10^4^/well) and L02 cells (8 × 10^3^/well) were exposed to different concentrations of WIP2W (1, 5, 10, 20, and 40 μM) for 24 h. PBS-treated K562 and L02 cells were used as controls (100%). Results are presented as mean ± SD (*n* = 6). The inserted graphic shows the WT1 expression levels of K562 and L02 cells. GAPDH served as the reference protein. (**B**) Cell cycle distribution of the K562 cells was determined using flow cytometry after treatment with 10 μM WIP2W for 24 h. Results are represented as mean ± SD (*n* = 3, *** *p* < 0.001). (**C**) Representative flow cytometry images of cell cycle assay.

**Figure 3 pharmaceutics-15-02305-f003:**
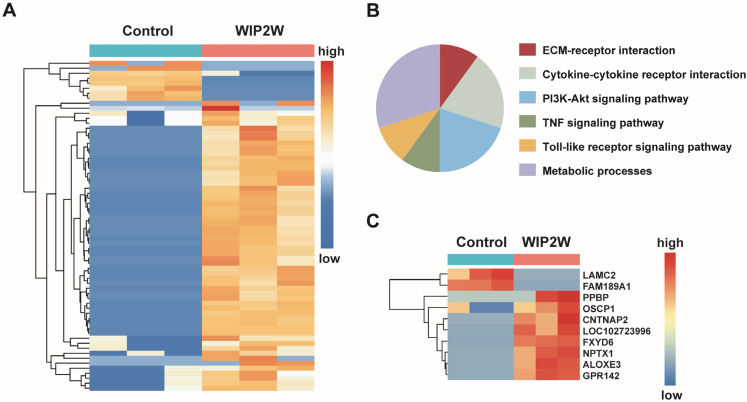
Gene expression changed in WIP2W-treated K562 cells. (**A**) Heatmap showing a hierarchical clustering of the differentially expressed genes in K562 cells after treatment with 10 μM WIP2W when compared to the untreated group (control). (**B**) Altered signaling pathway analysis of K562 cells after WIP2W treatment. (**C**) Selection of the potential genes in regulating WIP2W-induced K562 cell growth inhibition.

**Figure 4 pharmaceutics-15-02305-f004:**
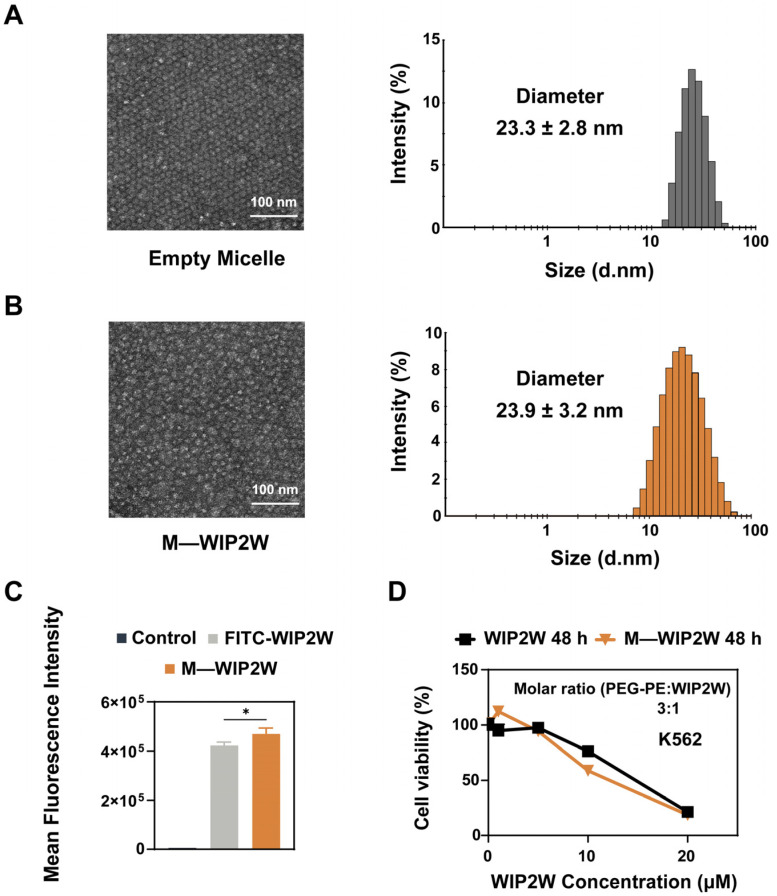
Characterization of empty PEG-PE micelles and WIP2W-encapsulating PEG-PE micelles (M—WIP2W). Particle sizes and morphology of (**A**) empty micelle and (**B**) M—WIP2W. (**C**) Mean fluorescence intensities of K562 cells after treatments with FITC-WIP2W (5 μM) and M—WIP2W (PEG-PE: 15 μM, FITC-WIP2W: 5 μM) in a complete medium for 2 h at 37 °C were determined using flow cytometry. * *p* < 0.05. (**D**) Cytotoxicity of WIP2W (1, 5, 10, and 20 μM) and M—WIP2W (WIP2W: 1, 5, 10, and 20 μM; PEG-PE: 3, 15, 30, and 60 μM) on K562 cells after 48 h of incubation at 37 °C, respectively.

**Figure 5 pharmaceutics-15-02305-f005:**
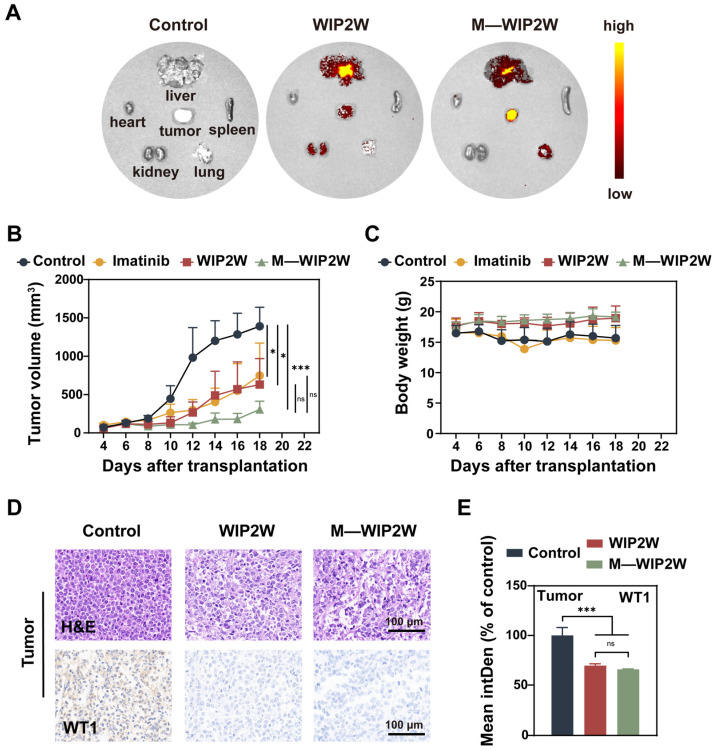
In vivo biodistribution and therapeutic effect of WIP2W and M—WIP2W in WT1^+^ BP-CML mice. (**A**) WIP2W distribution in the dissected tumors and major organs after treatment with FITC-WIP2W (30 mg/kg) and M—WIP2W (molar ratio of PEG-PE:FITC-WIP2W = 3:1) via IVIS spectrum, respectively. Tumor volumes (mm^3^) (**B**) and body weights (g) (**C**) of WT1^+^ BP-CML mice after different treatments with PBS (control), imatinib, WIP2W, and M—WIP2W were calculated every other day, respectively. * *p* < 0.05; *** *p* < 0.001; ns: No significance. (**D**) The excised tumors of WT1^+^ BP-CML mice after PBS, WIP2W, and M—WIP2W treatments, were stained with H&E and an anti-WT1 antibody (scale bar: 100 μm), respectively. (**E**) Statistical analysis of WT1 expression levels by measuring mean InterDen of WT1-stained areas from (**D**). The control group was set as 100%. *** *p* < 0.001; ns: No significance.

**Figure 6 pharmaceutics-15-02305-f006:**
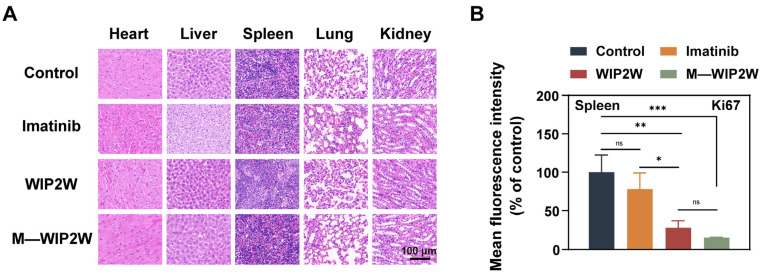
Biological safety and splenic infiltration evaluation of WIP2W, M—WIP2W, and imatinib. (**A**) Histological organ changes in WT1^+^ BP-CML mice after treatments with PBS (control), imatinib, WIP2W, and M—WIP2W, respectively (scale bar: 100 μm). (**B**) Statistical results of Ki67 fluorescent staining of spleen tissues after various treatments were obtained by measuring the fluorescence intensity from Appendix A. The data are presented as mean ± SD (*n* = 3). * *p* < 0.05; ** *p* < 0.01; *** *p* < 0.001; ns: No significance.

**Table 1 pharmaceutics-15-02305-t001:** Characterization of the empty micelle and M—WIP2W.

Name	Diameter (nm)	Zeta Potential (mV)	PDI
Empty Micelle	23.3 ± 2.8	−4.0 ± 0.6	0.188
M—WIP2W	23.9 ± 3.2	5.3 ± 0.3	0.253

## Data Availability

Not applicable.

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
