# Peer review of "A Nucleus-Targeting WT1 Antagonistic Peptide Encapsulated in Polymeric Nanomicelles Combats Refractory Chronic Myeloid Leukemia"

_pharmaceutics, 2023, doi:10.3390/pharmaceutics15092305_

Round 1
Reviewer 1 Report
The article entitled A nucleus-targeting WT1 antagonistic peptide encapsulated by 2 polymeric nanomicelles combat refractory chronic myeloid leukemia by Mengting Chen et al. discussed the role of WT1 inhibitor in controlling CML.
The article is well-written, and the data were appropriately discussed, however, I have some comments:
1. I would preferer if you add a list of abbreviations so that the reader can refer to during the journey especially that the manuscript is full of them.
2. A hint on the micelle and other related nanoparticles is needed in the instruction.
3. PEG-PE: do you mean Polyethylene glycol-phosphatidylethanolamine (PEG-PE)? Pls consider.
4. Line 105 and line 138: media not mediums
5. Line 106: could you justify using Female BALB/c nude mice?
6. Line 109: … expression levels in K562
7. Line 110: cultivated not cultivating cells.
8. Line 112: you might mean appropriate NOT desired
9. Line 142-145: paraphrase the paragraph
10. Line 147: 10 μM per what?
11. Line 149: where is the subject of the sentence? The Illumina NovaSeq 6000 platform sequenced the library, generating 150bp paired-end reads.
12. Line 160: conditions instead of circumstances
13. Line 193. Dot at the end.
14. Line 202: as follows.
15. Line 210: dote at the end.
16. Line 282-283: paraphrase the sentence to clarify.
17. Line 313: full stop
18. Line 388: nor need for “respectively”
19. Lines 476-79: pls clarify the sentence
General comments
1. In the majority of cases, you used inly the cancer cells and NOT the normal cells. Do you have data for these Normal cells? This is an example: “The uptake efficiency of WIP2W and M—WIP2W was reflected as the positive rate. Briefly, K562 cells were processed with FITC-WIP2W (0.05, 0.5, 5, and 7.5 μM) and M—WIP2W (PEG-PE: 22.5 μM, FITC-WIP2W: 7.5 μM) at 37 °C for 2 h. The positive rate (%) of these samples was determined using flow cytometer. The results are presented as N-fold 181 fluorescence intensity compared to untreated K562 cells.”
2. In the discussion section, a paragraph is needed to address the well-know or proposed molecular mechanisms of WT1 inhibitors in Leukemia.

Reviewer 2 Report
This is a review of the manuscript entitled " A nucleus-targeting WT1 antagonistic peptide encapsulated by polymeric nano micelle combat refractory chronic myeloid leukemia" submitted by Mengting Chen et al.
The manuscript is rich and meaningful, focusing on the efficient delivery of nucleus-targeting WT1 antagonistic peptide WIP2W encapsulated by nano micell. This system is a potent, selective WT1 inhibitor with noteworthy potency against CML.
This manuscript is well written and interesting. Therefore, I recommend this manuscript for publication after minor revision:
1) The WT1 inhibitory sequence was used on the N-terminal side, followed by the TAT sequence, but it is thought that a design with the TAT on the N-terminal side could have been used. Table S1 does not show a sequence where the TAT comes to the N-terminus, but did you do either in your pre-screening? If there is a rationale for such an order, please add it.
2)
The encapsulation efficiency and the uptake ability of WIP2W with and without nanomicelles were evaluated. but do you have any preliminary data on stability against proteases?
I wonder if the WIP2W peptide is distributed inside only. Is it possible that the physical properties of this peptide, especially its polarity as a molecule, would not allow it to exist near the surface of the micelle?
3) P5, lines 193 and 210
Please check the era because it has a strange shape.
4) P8, line 313 and Figure 3 line 319
There is no period at the end of the sentence.
5)
Does PEG-PE mean polyethylene glycol-phosphatidylethanolamine? must be specified once.
Reviewer 3 Report
Dear Authors
The article entitled “A nucleus-targeting WT1 antagonistic peptide encapsulated by polymeric nanomicelles combat refractory chronic myeloid leukemia” is an interesting research outcome. Following are the few suggestions that might can improve the article a bit
1. What is the significance of the micelles for the peptide delivery in this research, please add in the introduction section?
2. Novelty statement of this research is missing. Provide a clear novelty statement in the introduction section for better understanding
3. Provide the detail of the synthesis of the peptides and provide confirmatory evidence of the conjugation.
4. What is the pH of the PBS used for the micelle’s formation or any other study, clearly need to be mentioned throughout the articles?
5. What is the reason for the selection of the PEG-PE for the micelle’s formation?
6. Please provide a better resolution images of Figure 1
7. Provide the Full name of the PEG-PE in the article at least once.
8. Figure 4A and B need to be changed with higher resolution and clear particle size visibility
9. Rewrite the conclusion with better clearance to support the novelty of the project
Minor English editing is needed.
Round 2
Reviewer 3 Report
Dear Author
The article is well revised and good to go for the publication
Minor spelling check